# Basic Quality Controls Used in Skin Tissue Engineering

**DOI:** 10.3390/life11101033

**Published:** 2021-09-30

**Authors:** Laura Linares-Gonzalez, Teresa Rodenas-Herranz, Fernando Campos, Ricardo Ruiz-Villaverde, Víctor Carriel

**Affiliations:** 1Servicio de Dermatología, Hospital Universitario San Cecilio, 18016 Granada, Spain; laura.linares.sspa@juntadeandalucia.es (L.L.-G.); teresa.rodenas.sspa@juntadeandalucia.es (T.R.-H.); 2Ibs. GRANADA, Instituto Biosanitario de Granada, 18016 Granada, Spain; fcampos@ugr.es (F.C.); vcarriel@ugr.es (V.C.); 3Department of Histology, University of Granada, 18016 Granada, Spain

**Keywords:** skin tissue engineering, quality control, histological techniques, biochemical methods, biomechanical evaluation

## Abstract

Reconstruction of skin defects is often a challenging effort due to the currently limited reconstructive options. In this sense, tissue engineering has emerged as a possible alternative to replace or repair diseased or damaged tissues from the patient’s own cells. A substantial number of tissue-engineered skin substitutes (TESSs) have been conceived and evaluated in vitro and in vivo showing promising results in the preclinical stage. However, only a few constructs have been used in the clinic. The lack of standardization in evaluation methods employed may in part be responsible for this discrepancy. This review covers the most well-known and up-to-date methods for evaluating the optimization of new TESSs and orientative guidelines for the evaluation of TESSs are proposed.

## 1. Introduction

The skin is the largest organ of the body and performs many important physiological functions. This organ provides protection against a wide range of agents, acts as a sensory organ and plays key roles in the hydroelectrolytic balance and termoregulation [1]. The skin is frequently affected by several pathological conditions such as metabolic or genetic disorders, infectious diseases, primary or metastatic cancers, traumatic and burns injuries, etc. Some of these conditions can lead to severe structural defects or even skin loss and therefore the surgical repair in these patients is frequently needed.

Currently, the most common clinical solution for patients with large skin defects is the use of skin autograft (skin grafts obtained from healthy anatomic locations) or skin allografts (healthy skin grafts obtained from donors). These techniques have several drawbacks and limitations [2] and therefore more efficient alternatives are urgently needed. Consequently, the generation of tissue-engineered skin substitutes (TESSs) by tissue engineering (TE) emerged as a promising alternative for these patients. Some of these engineered skin models showed promising ex vivo, in vivo and even clinical results. However, as in other applications, technical improvements and optimization of these skin models are still necessary in order to elaborate more efficient, functional and biomimetic TESSs for these patients worldwide, a process in which quality controls of the products generated, play a crucial role.

The aim of this review is to provide conceptual and technical information concerning skin tissue engineering with special focus on the ex vivo and in vivo quality controls used to determine the potential clinical usefulness of TESSs. First, a general overview of the skin structure, function and regeneration is provided followed by the currently used surgical skin repair techniques. Secondly, the tissue engineering strategies used to generate skin substitutes will be discussed. Finally, a comprehensive review of the ex vivo and in vivo quality controls, which are frequently used to evaluate TESSs, are reported.

## 2. Biology of the Skin

The skin is composed by three major layers: epidermis, dermis and the subcutaneous tissue or hypodermis and appendages (Figure 1). Each layer has unique histological features and functions, with differences in function to gender, race, anatomical location and age [3].

As briefly mentioned above, the skin is a vital organ of our anatomy and their functions are extremely diverse. It is an efficient physical barrier and immunological organ against external harmful agents (physical, chemical and microbiological). Through its circulatory system, glands and rich innervation, the skin participates in body temperature and electrolyte regulation. Furthermore, the skin is essential for the sensuality and well-being of human beings [3].

The epidermis is strongly linked by hemidesmosomes to the underlying dermis through a macromolecular complex, the basal membrane. Histologically, the epidermis is a stratified epithelium mainly made up of ectoderm-derived keratinocytes, which represents between 90 to 95% of epidermal cells [1,3] and their cytoskeleton is made up of several cytokeratins (CK) [4,5]. These cells, due to their progressive differentiation process of approximately 28 to 30 days, are responsible for the polarity, stratification and keratinization of the epidermis [6]. Keratinocytes establish strong cell–cell and cell–basal membrane interactions which are necessary for preserving the cohesion and structure of the epidermis. Besides keratinocytes, the epidermis contains less abundant (5 to 10%) but functionally important nonectodermal cells such as melanocytes (derived from the neural crest), Langerhans dendritic cells (derived from the bone marrow), and Merkel cells (descended from epidermal linage) [3,7].

Melanocytes produce and transfer the melanin to the neighboring basal keratinocytes to provide protection against the damaging impact of solar UV radiation. Melanocytes can be immunohistochemically identified by Melan A, vimentin and S-100 proteins as well as by DOPA oxidase activity [8,9]. Langerhans cells, positive for CD 1a, are epidermal antigen-presenting cells and, they can recognize and process antigens present at this level and present them to naive T lymphocytes to initiate an immunological response. Merkel cells act as a type 1 mechano-receptors and are responsible of the sense of light touches and are positive for CK 20 [3,10].

In relation to the basal membrane (BM), it is a highly specialized thin sheet composed by a mix of fibrillar and nonfibrillar extracellular matrix (ECM) molecules that provides support to surrounding epithelia, muscle fibers, blood vessels and peripheral nerve fibers [11,12].

The keratinocytes are well-attached to the BM via hemidesmosomes and integrins being these epithelial-ECM interaction crucial for the normal function of the skin [11]. From the molecular point of view, the BM is complex and the most well-known molecules are the collagens type IV (main stabilizing molecule) and VII (anchor molecule), laminin (glycoprotein), nidogen/entactin (small glycoproteins) and perlecan (heparan sulfate proteoglycan) [11]. The integrity of the BM can be affected by some pathological disorders (Bullous pemphigoid) and it is histologically evaluated with some histochemical (PAS staining) and immunohistochemical stainings (laminin, collagen type IV). The fine ultrastructure of the BM can be well-evaluated by transmission electron microscopy (TEM) which confirm that it is composed by, from top to bottom, the lamina lucida, the lamina densa (reticular) and the sublamina dense [11,12,13].

Concerning the dermis, it is a dynamic supportive connective tissue which contributes to most of the skin’s mechanical support, rigidity, hydration and compression rate, and thickness [1]. The dermis is principally composed of a collagen-rich ECM in which diverse cells are embedded. Histologically, this layer can be divided in two zones, the papillary and reticular dermis. The first is recognized by a loose meshwork of thin and poorly organized collagens, mostly type III, and elastic fibers. The papillary dermis also contains nonfibrillar ECM molecules and a well-defined network of small blood vessels [14].

The reticular dermis is considerably thicker than papillary dermis, and it is mainly structured by well-organized thick bundles of collagens, mostly type I. These collagen-rich bundles are arranged parallel to the skin surface and in between run the elastic fibers. Just like in other tissues [15], the 3D organization of collagens depend on the presence of certain proteoglycans, such as decorin and versican [5].

The main and more important cells of the dermis are the fibroblasts. In physiological conditions the morphology of these cells is maintained by a vimentin cytoskeleton. However, under certain circumstances, such tissue repair, these cells express smooth muscle actin (SMA) and are known as myofibroblasts [1,3]. Fibroblasts are responsible of the synthesis and preservation of the dermal ECM and are crucial for tissue repair and regeneration [1,3]. Other cells, with immunological functions, which can be found in the dermis are mast cells (positive for metachromatic stainings and CD 117) and macrophages. In addition, the sebaceous and sweat glands, hair follicles, smooth muscle cells, blood vessels, peripheral nerves and nerve endings run through the dermis (Figure 1) [3]. Unfortunately, the development of these appendages are highly complex and their incorporation within TESS still represent a challenge in the field.

The subcutaneous cellular tissue is mostly composed of adipose tissue and joint the skin to other anatomical structures, such as muscles, cartilage, or bones. The tissue organization of the hypodermis differs greatly between individuals, gender, age and anatomical locations. In fact, it can be organized in small groups or lobules of adipose tissue surrounded by connective tissue septa (Figure 1). The hypodermis plays important roles in the thermoregulation process, insulation, nutrient storage and also provides an efficient mechanical support and protection of the body [3]. Moreover, this is the main source of adult adipose-derived mesenchymal stem cells, which are widely used in TE protocols.

## 3. Regeneration of the Skin

Multiple diseases and different kinds of traumatic injuries can affect the skin structure and function. One of the most important causes of significant skin loss is traumatic injury [16]. It is estimated that over 6000 people are hospitalized due to severe burns each year in Spain and approximately 200 of these patients unfortunately die [16]. On the other hand, surgical resection of different kinds of neoplasms could affect the structure of the skin and produces a broad spectrum of structural damage. Other reasons for skin loss are trauma and chronic ulcerations secondary to diabetes mellitus, pressure, and venous stasis [17,18].

The repair of this tissue system is carried out through a series of consecutive steps, among which we can mention: (a) performing hemostasis in order to seal the possible rupture. Its main purpose is to prevent infection; (b) tissue regeneration in order to replace those cells that have been lost in the process; (c) wound resolution with the objective of resolution of the different functions of the skin and remodeling of the new matrix [19].

After a structural damage of the skin, keratinocytes undergo a different process, known as keratinocyte activation cycle. Here, injured keratinocytes release IL-1, which triggers local blood vessel formation and immune response to the site of injury. This acute local inflammatory process carried out by keratinocytes, endothelial and immune cells jointly will eventually allow a partial closure of the wound [20]. Additional healing implicates the release a new basement membrane and TGF-β by fibroblasts. TGF-β plays an important role in this procedure shifting keratinocytes to the basal phenotype, suppressing hyperproliferation, activating the standard keratinocyte differentiation program, stimulating the production of extracellular matrix (ECM) components, and inducing normal keratinocyte stratification [21,22,23]. Unfortunately, some types of wounds fail to heal and require surgical interventions or bioactive dressing materials to stimulate and accelerate the healing process [2]. All these cellular and/or molecular processes are directly related to the histological and/or molecular quality controls discussed in the followed sections of this review.

## 4. Current Surgical Strategies for Skin Repair

After skin injury or loss, the preferred method for repair is the direct surgical repair. In the case of severe damage with large loss of substance there are others surgical options (Table 1). Flap surgery is a usual reconstructive technique that implicates moving healthy tissue (skin, fat or even muscle) from one location of the body to another adjacent damaged area. This flap generally remains partly connected to the body and its blood supply via a pedicle [24,25]. For this reason, wounds often cure without severe complications and the final cosmetic result is usually satisfactory (Figure 2). Another repair option is a skin graft. It is a surgical technique that allows a definitive coverage of wounds by transplanting tissue from one unaffected area of the body. It is important to consider that the effectiveness of these treatments, like in other disciplines [26,27], will depend on its structural and pro-regenerative properties. An ideal skin graft does not yet exist, but it must meet some important criteria, such as: i) being able to control infection; ii) avoiding fluid and temperature loss; iii) exhibiting adequate histocompatibility; iv) having a good and stable adhesion to the wound site; v) being able to respond adequately to anatomical and biomechanical needs; vi) not being toxic or triggering an immune response; vii) supporting epithelial and stromal regeneration and differentiation; viii) being cost-effective; ix) being instantly available for clinical use [28]. Autograft dressing (either meshed or unmeshed) is a graft obtained in the same individual and it is considered the best available technique for wound closure [29], which is able to provide skin integrity (cells and ECM) without rejection risk (Figure 3). However, this method is not always available and has several well-known disadvantages, such as creation a second surgery injury, restricted availability, significant contraction, shrinkage and scarring [30,31]. A method to overcome the limitations of the use of skin autograft are the use of skin allografts. They can be obtained from cadavers or living donors and are often used as a temporary prevention of wound contamination and fluid loss [32]. Unfortunately, this technique has the disadvantage that not enough tissue is available, due to large demand worldwide, few banks to collect and store these grafts, and many safety and ethical issues. Moreover, even after standardized sterilization and rigorous screening quality controls of these grafts, the transmission of viral diseases has not been completely eliminated [33]. Finally, the use of acellular xenogeneic dermal-derived grafts is being explored, but more research is still needed to elucidate their therapeutic efficacy. Where a lost skin area is unable to be repaired using these conventional surgical strategies, tissue engineered skin substitutes contribute promoting wound healing. Composite synthetic or biological dressings are often used to speed wound healing in chronic or burn wounds but they do not offer permanent treatment [24,25] and subsequent surgical interventions are often needed. These composites are commercially available in different sizes for use in skin repair. In general, these matrices are generated from highly purified bovine or pig ECM molecules, such as collagen or elastin, providing high levels of biocompatibility and degradability (Figure 4).

## 5. Skin Tissue Engineering

During recent years, several technological advances in TE field have been made, allowing new functional and clinically efficient TESSs to repair critical skin defects. The TE is a relatively novel scientific area that combines the principles and methods of engineering with biological structural bases, such as histology, with the aim to generate biologically functional engineered substitutes for the repair or replacement of injured human tissues or organs [34,35,36].

To search an efficient skin substitute to be used in the replacement of current grafts has spurred a number of researchers over the past 30 years. The closest early attempt to develop what might be called a synthetic skin or skin substitute was devised by Curtis (1951). This was fabricated from a gel composed by partially hydrolyzed casein, sodium lactate and sodium lauryl sulfate [37]. Since Green and Rheinwald’s (1975) technical contribution, which described how to isolate and subculture a large amount of human keratinocytes [38], several clinical trials tested the efficacy of keratinocyte-based sheets in the treatment of large skin loss [39,40]. Autologous epidermal sheets served as permanent wound coverage with reasonable cosmetic result and without any risk of rejection. However, this method has disadvantages such as needing 2–3 weeks to obtain a sufficient number of cells and high cost [41]. Furthermore, graft-take depends on several factors like wound preparation, intrinsic status, patient underlying diseases, and operator expertise [7]. Initially, some success was reported, but these grafts were never as good as the use of the standard split-thickness skin, probably due to the lack of a dermal cells and ECM components [42]. The allografts have the advantages to produce a temporary prevention of dehydration and contamination, promoting angiogenesis and less pain. Nevertheless their availability is also limited and an immune rejection and inflammation at the wound site may be important disadvantages.

In this regard, a wide range of synthetic-based engineered analogues to promote keratinocyte growth have been studied in animals and humans [12,43,44,45]. Furthermore, these analogues can be seeded with cells or be completely acellular, and both systems have been widely used in skin TE [46,47]. Some of the earliest versions have been the xenogenic composites made of purified bovine collagen and those generated with shark cartilage-derived chondroitin-6-sulfate, both with an outer silicone covering. Significant progress has also been made with the decellularization technique which allows to be generated natural and tissue specific acellular matrices for biomedical use [48], including skin repair. Through chemical, physical or biological procedures it is possible to remove the cellular contents from the tissues efficiently reducing the immunogenicity and thus reduce the risk of graft rejection [48,49]. However, the risk of viral or DNA traces transmission is small but not inconsiderable, even when sourced from accredited skin banks. In this context, peracetic acid, a widely used chemical decellularization agent, has the promising ability to get rid of such viral contents [50]. These decellularized skin allografts have been effectively used alone or in combination with cultured autologous keratinocytes [51].

The treatment of large and deep skin injuries is still a challenge in surgical dermatology, and researchers worldwide have been working on the development of full-thickness TESSs [2]. These models are generated through the combination of dermal and epidermal cells with an artificial matrix or scaffolds, often composed by collagen [52]. These engineered substitutes, mainly composed by a biomaterial containing dermal fibroblasts and keratinocytes showed promising morphofunctional properties like signs of epidermal differentiation, the establishment of dermoepidermal junctions, and the synthesis of some ECM molecules [2,53].

Regarding the biofabrication of bilayered skin constructs, the most common approach is the use of a hydrogel. It serves as a physical platform for dermal fibroblasts growth, on top of which keratinocytes are seeded and allowed to form an epidermal-like layer and mature in vitro [54]. To generate these substitutes a small healthy skin biopsy from the patient is required which is subsequently used to isolate and expand the dermal fibroblast and keratinocytes [54]. Each cell type must be expanded with specific culture medium, for which in the case of human keratinocytes the epidermal growth factor (EGF) and molecular cues—adhesive glycoproteins, functionalized culture flask or irradiated cell-based feeder layer, etc.—are needed to promote efficient growth and expansion [55]. Different skin substitutes have been generated, and in most of the cases dermal fibroblast are usually encapsulated or seeded on top a hydrogel to create an engineered stroma that has the aim to structurally and biologically mimics the dermis [56,57]. Most of the skin substitutes available use collagen type I as 3D scaffold. This is the main ECM molecule of the dermis, and collagen-based hydrogels are widely used in TE. However, it is important to take into account that the collagen used is xenogeneic (mostly bovine), tends to shrink, is rapidly biodegraded in vivo, and has poor biomechanical properties [24]. For these reasons, other biomaterials have been used in order to generate TESS with a higher level of efficiency and structurally more stable, such as collagen-chitosan/fibrin glue [58] (its main purpose is the proliferation and migration of fibroblasts, it has been difficult for the cells to epidermals grow and reach confluence), collagen-GAG [59] sponges (Collagen-GAG and collagen funnel-shaped collagen sponges have an upper surface layer and a bulk porous layer and in this way allow cells to dislodge homogeneously and improve cell viability). The incorporation of GAG increased the mechanical property and cell viability of collagen sponges, gelatin [60], thiol-norbornene cross-linked pectin hydrogels 202 decellularized dermis [61], fibrin [62], scaffolds based on synthetic biomaterial [63] (these scaffolds were able to support the growth of different cell types, including keratinocytes, fibroblasts, and endothelial cells, as well as the production of extracellular matrix, which ultimately leads to the production of new collagen), a self-assembly technique [64] or hybrid natural hydrogels composed of human fibrin and a small amount agarose [65]. Concerning the cell sources used in skin TE, most studies demonstrated that keratinocytes and fibroblast are suitable for the generation of functional and proregenerative TESSs. However, these cells are not always available and keratinocytes are difficult to expand in culture. For these reasons, researchers started to explore alternative cell sources for skin TE. In this sense, mesenchymal stem cells, which can be obtained in higher amount and has the capability to differentiate in different linages, emerged as a promising alternative for stem cell-based therapies in dermatology [66] and skin TE. These cells have been used as an alternative epithelial cell source for generation of bioengineered human skin substitutes with possible practical utility [67,68,69] and seem to improve skin regeneration on its own or in combination with a scaffold [70]. Although this different kinds of stem cell, currently used in a wide range of TE protocols, is a promising alternative in this field more research is required to demonstrate their potential clinical usefulness. More details about the use of stem cells in skin TE can be found in other review articles [71,72,73].

In order to generate a stratified epidermal layer, thus an efficient barrier, the TESSs are first fully immersed into the culture media for approximately 15 days. This step supports the formation of a monolayer of keratinocytes on top. After this period, TESSs are subjected, to another 15 days, to an air–liquid culture technique to induce a correct stratification and maturation of the engineered epidermis [24] (Figure 5). This well-established practice was successfully applied to generate bilayered skin grafts based on woven/non-woven fibers, porous freeze-dried scaffold, nanofibrous matrix, and even 3D printed hydrogels [54]. A schematic representation of the ex vivo and in vivo main features of the human fibrin-agarose skin substitute is shown in the Figure 5. This human skin model is being clinically evaluated in Andalusia, Spain [65].

These 3D models represent an important technical advance which solve the problems associated to the use of cell-sheets technique or 2D skin cell cultures. Indeed, the new generation of TESSs has demonstrated that they promote the recapitulation ex vivo of several morphofunctional features of native skin, such as barrier function, biomechanical resilience, keratinocyte stratification, the establishment of efficient cell–cell molecular interactions, and the synthesis of basal membrane and essential ECM molecules (collagens, glycoproteins and proteoglycans). On the other hand, a three-layer TESSs comprising a hypodermis better reproduced the skin histoarchitecture [74]. In addition, recent essays have been performed to develop an ideal skin substitute that accelerates wound healing via fast development of new vessels [29], the inclusion of adipose tissue-derived microvascular fragments [75], induced pluripotent stem cell (iPSC)-derived endothelial cells into the dermal component [76] and the incorporation of antimicrobial properties by using antibiotic-loaded nanoparticles [48]. However, it is still a challenge to generate TESSs that has immunological properties, an adequate degree of pigmentation, a predefined vascular network and the presence of glands and/or hair follicles precursors.

During the last years the three dimensional (3D) printing technology was applied to generate TESSs with success. It uses a computer-aided design to fabricate functional tissues and organs via a layer-by-layer positioning of biomaterials and living cells [77,78]. It is an efficient alternative to emulate the complex histological 3D structure of the native skin [79]. This methodology was used to combine gelatin methacryloyl and alginate hydrogels to generate and subsequently incorporate layers of endothelial cells [80] or even microchannels [81] into a multilayered skin construct. Currently, several groups are introducing/optimizing the 3D printing technology for the generation of more biomimetic and efficient skin equivalent with promising preclinical results. However, more research is still needed to be able to transfer this methodology to the clinical practice [82,83].

## 6. Quality Controls in Skin Tissue Engineering

Based on the wide range of TESSs models developed within the last years, it is very important to correctly demonstrate that these substitutes are suitable for their use in skin repair. All these aspects make quality control methods an indispensable part of TE procedures.

Most TESSs are first evaluated ex vivo and, based on these findings, some of these substitutes could prove suitable for further in vivo preclinical evaluation [84,85]. Currently, there is a wide range of methods available to characterize TESSs generated ex vivo and tested in vivo, which include functional analyses, histology (light and electron microscopy), molecular biology and biomechanical testing [86,87]. Furthermore, the performance of all these preclinical studies is a prerequisite to obtain the approval by the specialized governmental agencies for future clinical use of the TESSs [88].

In the following sections a comprehensive review of the ex vivo and in vivo quality control methods used to characterize engineering TESSs is provided.

## 7. Ex Vivo Quality Controls

The ex vivo characterization of TESSs [89], such as in other biomedical applications [90], should demonstrate that the TESSs generated are composed by viable and functional cells. Furthermore, the combination of these cells with an adequate biomaterial should allow to be reproduced the main histological structure and some of the key functions of the skin (Table 2).

### 7.1. Assessment of the Cell Viability and Functionality

The assessment of the cell viability, often defined as the quantitative and/or qualitative determination of the number, percentage or fraction of viable and functional cells included in a determined cell culture or engineered substitute. Therefore, it is an important step to confirm that the engineered tissues generated are predominant composed by viable and functional cells [118]. In addition, it is also crucial to confirm the biocompatibility or cytotoxicity of the biomaterials used. From the technical point of view, the methods used can be exactly the same, but they are applied in different experimental contexts [119].

One of the most used methods for assessing cell viability was the dye exclusion method which use the Trypan blue azo dye [120,121,122]. It is based on the principle that viable cells, with an undamaged and functional cell membrane, exclude the dye. However, dead cells or those with irreversible cell-membrane damage (permeable cells) incorporate the dye molecules, and appear stained [120]. This method is mostly carried out on a counting chamber device with a gridded area which allows to be determined the concentration of cells in a determined volume and calculate the cell viability index. These devices are known as haemocytometer or Neubauer chamber slides.

Over time, the cell viability assays have become more complex. Indeed, dye-based colorimetric reactions that depend on the metabolic activity of the cells are gaining favor. These reagents can be applied to determine the metabolic activity of cells in suspension, attached to culture flasks or even those immersed or cultured on top of biomaterials [123]. The first, and probably best known, metabolic dye used for this purpose is the 3- (4,5-dimethylthiazol-2-yl)-2,5-diphenyltetrazolium bromide (MTT). MTT measures mitochondrial metabolism, which is an index of cellular viability rather than proliferation [124]. Similar reagents which allow to spectrophotometrically determine the enzymatic activity of the cells through a colorimetric reaction are the commercially available WST-1 (water-soluble tetrazolium salt) and PrestoBlue^®^ assays (Thermo Fisher Scientific, Waltham, MA, USA). WST-1 have been often used to determine the cell viability in TE [125], but also the biocompatibility or cytotoxicity of different biomaterials [126].

On the other hand, fluorescein has also been employed in TE to assess the in vitro viability of a variety of cell types and even tissues, including human keratinocytes [100]. This dye was also clinically used to assess skin flap viability [127]. Based on its bipolar side chains, the fluorescein diacetate (FdA) can easily penetrate the cell membrane and is degraded by intracellular esterase’s, releasing the fluorescein which remains intracellular in viable cells with intact membranes, allowing the identification of these cells under fluorescence microscopy [120]. This method provides a global, nondestructive measure of the distribution of viable cells within skin constructs [100]. Similarly, live/dead viability staining describes a mixture of two fluorescent dyes that differentially identify live (usually resulting in a green fluorescence) and dead cells (usually with a red fluorescence) [128].

There are few other fluorochromes which can cross the cell membrane of live intact cells, such as Hoechts (HO33342). This fluorescent dye is one of the most popular supravital probe which binds to the DNA. When this reagent is used at low concentration, it will allow the identification of intact or viable cells, which metabolically introduce lower concentrations of the dye, by a bluish fluorescence or damaged/dead cells which exhibit a metachromatic white-yellow emission, due to the high concentration of the dye [129]. The combination of HO with propidium iodide allows to obtain more accurate information of the cell viability [130]. The supravital fluorochrome-based stainings can be used to determine the cell viability and functionality of cells in suspension or cultured within culture flask or biomaterials. However, these dyes can provide reliable quantitative results of cells in suspension when analyzed by flow cytometry [130].

Finally, determination of cell viability by electron-probe X-ray microanalysis (EPXMA) is also used for identifying, localizing, and quantifying the ionic elemental composition related to cell viability, functionality or death (such as potassium, sulfur, sodium, calcium, chlorine, phosphorous and magnesium). This method allows to determine the concentration of these ions in a whole cell or at the intracellular or even organelle levels [131,132]. This is probably the most specific and sensible method available to accurately determine the cell viability [125]. This technique is frequently used within conventional cell cultures, but it can also be applied to evaluate the biocompatibility of biomaterials in TE [133]. However, it is an expensive technique which requires a scanning electron microscope equipped with an EPXMA, which limits it use [122].

### 7.2. Histological Assessment

Skin histology is considered one of the most solid quality control methods to evaluate skin regeneration and/or pathological processes [86]. Histological analyses can be performed on either frozen sections or paraffin-embedded tissues, being the paraffin-embedded tissues the most commonly used [134]. A wide range of diverse histological staining methods are available depending on the structure of interest to be studied [134].

The histological analysis of TESSs must be oriented to demonstrate the establishment of a well-structured and functional epidermis. In the case of the stroma, it will be necessary to demonstrate the homogeneous distribution of functional and viable cells. Stromal cells should be able to positively interact with the biomaterial showing an adequate elongated morphology, proliferation rate and/or ECM synthesis.

The starting point for histological analyses is the haematoxylin-eosin (HE) routine staining. This method allows the staining of cellular and tissue elements through the use of a basic (haematoxylin) and an acid (eosin) dye. This simple method allows to evaluate the general overview of native or engineered models histoarchitecture [12,135] confirming the establishment of an engineered epidermis and dermis. However, HE staining is not an accurate and specific method to evaluate other essential cell or tissue elements, such as the ECM molecular composition or the expression profile of tissue specific or functional proteins (differentiation or cell linage markers) [12]. Therefore, histochemical and immunohistochemical methods should be used.

In general, most of the TESSs have demonstrated the establishment of a relatively well-stratified epithelium composed by viable and functional keratinocytes layers and a new-formed basal membrane. Some of these models achieve a closely similar histological pattern than a healthy human epidermis [94,97,136,137]. In order to specifically determine the expression of epithelial, dermal or ECM molecules in TESSs, immunohistochemical procedures are used.

As mentioned above, keratinocytes are the main structural cells of the epidermis, they initiate a progressive differentiation process from the basal layer to the superficial stratum corneum. Throughout this process, keratinocytes modify their morphology and the expression of their main cytoskeletal intermediate filaments, the cytokeratins (CKs). There are two groups, acid or type I and neutral-basic or type II CKs. These cytoskeletal proteins form heterodimers composed by one of each group [4] and they show a differential expression pattern according to the epidermal layer and/or skin anatomical location (Table 3).

In the case of the stroma, some authors investigated the functional properties of encapsulated fibroblast (or other cell sources) [164]. To evaluate the engineered dermis some classic histochemical methods can be useful tools. In this sense, fibrillar collagens, the most abundant fibers of the dermis (collagen type I and III), can be identified by histochemical methods, such as Masson, Van Gieson, picrosirius or the integrated histochemical approach called Fontana–Masson picrosirius (FMPS) [8]. Interestingly, these methods can also be applied to evaluate the structural organization of collagen-based TESSs or decellularized matrices. In this context, picrosirius-based techniques increase the natural birefringence of collagen fibers allowing us to determine the collagen organization pattern under polarized light microscopy [8]. Concerning the elastic fibers, they can be easily and specifically identified by orcein, aldehyde fuchsine or Verhoeff stainings. The reticular fibers, which can be observed in the skin of some anatomical locations, can be well-identified by using metal reduction (such as Gomori, Lynch reticulin or Gordon and Sweet methods) or PAS (periodic-acid Schiff) techniques. Both methods demonstrate the carbohydrates associated with these fibers. Based on this basic principle, PAS histochemical method is also a useful alternative to demonstrate glycoproteins, especially those which form part of basal membrane. Although for the evaluation of the basal membrane there exists a number of techniques, such as methenamine-silver technique, it is better to use specific immunohistochemical markers [3].

Regarding the dermal proteoglycans, there are several types in the skin. Proteoglycans play key roles during collagen fibrillogenesis, regulating the cell function (storing or presenting growth factors), providing an adequate hydration rate to the stroma, and participating in the wound healing process [19]. Histochemistry can help to determine the synthesis and deposition of proteoglycans in TESSs ex vivo or even to confirm their presence when they were used as scaffold. The methods available for this propose are the alcian blue (pH 1, 2.5 or 4), safranin O, toluidine blue and many other stainings [5,9,65]. These techniques are sensible enough to provide a general, but not specific, overview of the presence of these molecules, that could be part of the biomaterials used or be produced by the cells within the engineered tissues [24]. However, further molecular-based staining, such as immunohistochemistry, are needed to determine the presence and distribution of specific fibrillar or nonfibrillar ECM molecules.

In this context, immunohistochemistry will allow to accurately demonstrate the synthesis and distribution pattern of ECM molecules within the structural framework of the TESSs (Table 4). To demonstrate the synthesis of a basal membrane ex vivo, the immunohistochemical identification of laminin isoforms and collagen type IV are useful and highly specific alternatives. Furthermore, the identification of integrin α6β4 will help to demonstrate if the basal keratinocytes within engineered epidermis were able or not to establish cell–basal membrane interactions [104]. Integrins are the main adhesion proteins that communicate the cellular cytoskeleton with the ECM. These molecules are bidirectional signal transducers which regulate cell proliferation, differentiation, adhesion and migration. Therefore, integrins are fundamental for keratinocytes migration during development, regeneration and physiological functions [104].

The analysis of the CKs expression patterns in TE can provide solid evidence about the stratification and maturation of ketatinocytes in TESSs. These analyses are essential to confirm that the engineered epidermis have a similar structural pattern and CKs expression than native and functional human epidermis [77]. Other proteins which are frequently used as epithelial differentiation markers are filaggrin, involucrin [67], loricrin or transglutaminase [53]. These proteins, as described in previous sections, are part of the barrier that protect the internal environment from external harmful agents and minimizing the loss of water and other fundamental components to the outside. Additionally, immunohistochemistry is a valuable option to demonstrate the establishment of different kind of cell-cell interactions, which are important for the barrier function of the epidermis, like desmoplakin, plakoglobin or plakophilins (Table 3). Finally, the cell proliferation of the keratinocytes or fibroblast can be easily assessed by using antibodies against Ki-67, PCNA or cyclins. The quantitative analysis of these proteins allow to define the cell proliferation index, a practical indicator of cell viability [12]. Furthermore, apoptosis can be evaluated by immunohistochemistry (caspases) or TUNEL assay.

Concerning the ECM of the dermis within engineered TESSs, it is probably that fibroblasts are not able to produce a high amount of diverse ECM molecules under ex vivo conditions. Histochemistry could provide some results, but the immunohistochemical analysis of most abundant ECM molecules will help to confirm the normal fibroblast function. In this context, the immunohistochemistry for the identification of the main and more abundant ECM molecules will demonstrate the ability of fibroblast to synthesize ECM molecules for example collagens (type I, III and V), glycoproteins (fibronectin) and proteoglycans (decorin, biglycan, versican, etc.). In relation to the identification of elastic and reticular fibers, histochemical methods are more sensitive, easier, faster and cheaper alternatives than immunohistochemistry. However, the immunohistochemistry will allow to be determined the specific expression of the microfibrils, glycoproteins which form part of the elastic fibers, such as fibrillin I and/or II [174].

The fibroblast can be immunohistochemically identified by using antibodies against their intermediate filament vimentin. When fibroblast acquires a reparative phenotype, often observed during wound healing process, expressed contractile cytoskeletal proteins such as a-smooth muscle actin allowing to identify them by immunohistochemistry. In some TESSs a combination of cells are used, especially within the stroma. In this context, the distribution of endothelial or endothelial-like cells can be assessed by the identification of CD-31 or clotting factor VIII.

### 7.3. Transmission Electron Microscopy

In addition to conventional light microscopy, transmission electron microscopy (TEM) may be used to evaluate the ultrastructural features of TESSs [51]. This kind of microscopy will allow us to clearly identify pathognomonic features at the epidermal, stromal, and ECM levels. In the case of the keratinocytes, TEM will allow to confirm their stratification features, such as the progressive establishment and enhance the cytoskeletal CKs organization and the interdigitated cell–cell interactions. Indeed, TEM is the most accurate method to evaluate the features of the cell–cell desmosomal interactions with their typical thick bundles of CKs filaments, dense plaques and outer and inner zones [178]. In relation to the basal membrane, if epithelial and stromal cells formed a basal membrane ex vivo, TEM will help to reveal the ultrastructural features. In this sense, it will be possible to distinguish the two main layers: the basal lamina or lamina basalis and the reticular lamina or fibroreticularis and the presence of the hemidesmosomes. A mature and well-structured epithelial lamina basalis should be composed by the lamina rara, a narrow space between the cells and the second element the lamina densa [178,179].

At the stromal level, TEM will provide clear images of the main fibers of the ECM molecules, such as the collagen fibers, with their characteristic striated pattern, and elastic fibers, with their elastin protein core and surrounding microfibrils [178]. In relation to the fibroblast, TEM will show their main organelles (packed rough endoplasmic reticulum, Golgi apparatus, and a prominent nuclei and nucleoli) and shape (elongated cells with multiple surface folds and exocytosis vesicles) [178]. All these ultrastructural features are critical for the stability of the dermal–epidermal junction and barrier function.

In general, it is recommended to conduct several techniques to demonstrate accurately the histological and ultrastructural features of engineered tissues ex vivo. The use of one method will provide limited information, for example, it was immunohistochemically demonstrated that mesenchymal stem cells cultured within biomaterials can proliferate and produce diverse ECM, like collagens or laminin ex vivo. However, TEM analysis did not demonstrate the existence of mature and well-organized collagen fibers within the designed tissues [180]. Immunohistochemistry is a reliable and useful method, but it is not uncommon for us to resort to TEM to confirm these findings.

Another emerging technique with some appeal but much less frequently used to perform high-resolution images of tissue structure on the micron scale is the optical coherence tomography (OCT) system [110]. OCT is a relatively recent noninvasive and nondestructive optical imaging technique based on measuring backscattered or backreflected light. OCT generates 2D and 3D tomographic images with a micron resolution. The results show a high degree of correlation with the histological findings regarding structure and layer thicknesses. Imaging can be performed in situ, without removing a tissue specimen and in real time [108]. OCT has been increasingly used in dermatology [181,182,183] for the evaluation of wound healing [184] and other varied fields, including the diagnosis of melanoma [168]. Unfortunately, other smaller and highly specific features, such as the epidermal–dermal junction and cellular features, cannot currently be visualized [185]. More research is needed to determine the usefulness of the OCT in skin TE ex vivo and in vivo.

### 7.4. Molecular Biology

ECM components and DNA content can be quantified through a variety of as stated [186]. These techniques make it possible to complete all the semiquantitative information obtained by the histological examination and help to identify similarities and deficiencies between the tissue that has been designed and the native tissues [107]. For example, the participation of fibroblasts in the formation of the dermoepidermal junction has been confirmed by the reverse transcription polymerase chain reaction (RT-PCR). This technique shows dynamic interactions between fibroblasts and keratinocytes during in vitro maturation, as well as the marked changes that occur after in vivo transplantation [70,117]. Furthermore, semiquantitative analysis of specific epithelial or stromal proteins, used as markers for the identification of cells and ECM molecules, can be easily performed by Western blot. Another useful technique is the determination of gene expression profile by microarray technology, which allows to evaluate several genes simultaneously and quantitatively. These findings can orient prospective designs of TESSs for more physiologic characteristics.

The highly specific information provided by the molecular biology is useful to confirm the gene or protein profile of TESSs as compared to native skin or pathological conditions. However, these methods should be used synergistically to support, confirm and complete the histological findings, but not as the unique quality control method in TE, since the histological pattern of the generated substitute must always be demonstrated.

### 7.5. Biomechanical Characterization

An ideal TESS should be identical to normal human skin, not only biologically, but also biomechanically [173]. Nowadays, overall biomechanical properties of any engineered construct can be measured through the application of different forces under specific conditions (Table 5). In this context, conventional techniques to evaluate the biomechanical functionality of artificial skin substitutes include tensile, compression, and shear stress tests [78,116].

The most common method of loading for characterizing materials is uniaxial loading by tensile test. For the tensile test, the specimens are aligned with their longer length in a parallel direction to the tensile force (Figure 6). In order to be accurate with the measurements and the results obtained, a series of variables must be taken under consideration. A fundamental variable is the distance between the clamps that must be constant and a constant strain should be applied. Another criteria to be considered is the shape and size of the samples since they can affect the results of the tensile test [198]. The results of the uniaxial tests are plotted on a stress-strain curve, which is represented by the response of the material to applied forces. From this curve, important information on the material’s load capacity can be obtained and we can differentiate three regions: the elastic region, in which the material returns to the nondeformed state when forces are applied; the plastic region, in which the material deforms permanently, and the failure region where the tensile strength is reached and the fibers begin to break sequentially until they all break completely.

In addition, there are parameters, such as Young’s modulus, maximum strength or tensile strength, and elongation at break or strain at break, which are calculated from the stress-strain curve after the tensile test. Young’s modulus (E) is the most common parameter since it indicates how rigid the material is and it is calculated as the tangent modulus of the initial linear portion of the stress-strain curve of each sample. The ultimate strength or tensile strength (break σ) has as its ultimate goal the measurement of the maximum stress that a plastic sample can withstand while being stretched before breaking. This can be calculated using the following formula: σ (stress) = F/A, where F: normal force acting perpendicular to area and A: area. Finally, the elongation at break or deformation at break (ε-break) is the ratio between the increase in length and the initial length after failure of the test specimen. Elongation is calculated as the relative increase in length. ɛ = (ΔL/L) x 100, where ΔL: final length and L: initial length.

### 7.6. Functional Evaluation

One critical function of skin is to form an effective barrier to protect the body from penetration of infectious agents and loss of water and necessary nutrients. The skin barrier function can be measured using skin biophysical instrumentation by noninvasive methods (surface hydration [93,96,199] transepidermal water loss [102,200] and invasive methods (water permeation [201,202], niacinamide flux [201] in vitro and in vivo [145]).

Surface hydration: human stratum corneum electrical conductance depend on their water content. This is the reason why capacitance and/or conductance methods are commonly used to measure water content of TESSs and development of barrier function in vivo.Transepidermal water loss (TEWL). TEWL is the amount of water vapor evaporating from a fixed surface of the skin per unit time. It is measured using sensors that detect changes in water vapor density. Niacinamide flux: Permeability of niacinamide has been revealed as a sensitive invasive method to measure the barrier function in cultured skin substitutes.

Although restoration skin barrier is essential for the general well-being of the body, this function was not often measured directly in most reviewed articles.

## 8. In Vivo Quality Controls

In vivo studies are crucial and must be focused on demonstrating the therapeutic efficacy or failure of the use of TESSs in models of skin repair (Table 3). If positive results were obtained, the TESSs should support a proper epithelial maturation with a collagen-rich and vascularized stroma. Furthermore, these quality controls should help to accurately elucidate if the TESSs were biointegrated, biodegraded, encapsulated by the host connective tissue, or rejected by the host immunological system.

### 8.1. Macroscopic Evaluation

Macroscopic aspects of the wound healing process are important because they provide information related to the success or failure of the graft used. Some important aspects include the TESSs take rates, color (pigmentation, and signs of vascularization), elasticity, surface smoothness, if results are comparable to the surrounding skin. Furthermore, these evaluations should confirm the absence of signs of infection, contraction, inflammation or necrosis [97,128,136,145,160]. While it is true that most studies reviewed supply a macroscopic description of the wound healing process, these descriptions remain unsystematic. For this reason, the generalized use of a standard checklist would improve the quality of macroscopic evaluation and the comparison between studies [120].

### 8.2. Histological and Ultrastructural Analyses

Almost all studies reviewed employed routine histological analyses to determine the efficacy of TESSs in skin repair. In general, HE staining is employed for general evaluation of the histological pattern and host tissue response to the grafted engineered tissues. This method will confirm the epidermal growth (implanted cells and/or from the host), the degree of stratification as well as some undesired results such as partial epithelization, epithelial lost and/or inflammatory infiltration. In the case of the stroma, this routinely used method provides useful information concerning the presence and thickness of granulation tissue, the fate of the biomaterial (biodegradation, integration, encapsulation or rejection), extracellular matrix remodeling, degree of vascularization, and if newly formed rete ridges, papillary plexus, and appendages have started to be regenerated.

HE staining also allows the identification and localization of different cell types involved in the wound healing or inflammatory processes, like macrophages, granulocytes (neutrophils, eosinophils, basophils), lymphocytes, plasma cells, foreign body reaction multinucleated giant cells, and fibroblasts [46,59,97,128,145,160]. However, it will not allow to properly identify the lymphocytes (B, T (helper CD4+ and cytotoxic CD8+), natural killers (NKs)) or macrophages types (M1 or M2). The adequate interpretation of HE and immunohistochemical markers will allow to be determined an innate or acute immunological response mediated by neutrophils, macrophages, mast cells and NKs; an adaptive or chronic response from lymphocytes [T and B] and plasma cells; or foreign body reaction of mononuclear cells, macrophages and giant cells to TESSs [48,177,203,204,205,206]. In addition, it is well-known that macrophages can acquire functionally distinct phenotypes, M1 and M2 macrophages. In general, M1 macrophages (express CD80, IFNγ, IL-6, IL-1β, MCP-1, iNOS, TNFα) exhibit strong proinflammatory properties whereas the M2 macrophages (express CD23, CD163, CD206, IL-10, IL-4R, TFGβ) appear to suppress immune surveillance promoting neovascularization [51,204,205].

Alternative methods, such as Masson’s trichrome [116,128], picrosirius, PAS and orcein stain [156] were used for more detailed distinction, between different tissue/biomaterial components [65,152,153]. These histochemical methods allow to confirm the progressive synthesis, remodeling of the host and/or newly-formed ECM of the dermis. Picrosirius under polarized microscopy will allow to determine the degree of organization and parallelism of the collagen network. Furthermore, these techniques may help to determine if an abnormal synthesis or fibrotic response to the grafted tissues occur, often associated to synthetic and/or slowly degrading biomaterials [207]. In a regenerative microenvironment, the fibroblast which expressed vimentin, can acquire contractile properties expression a-smooth muscle actin (SMA), which allows to identify them.

In addition with the assessment of the histological pattern, specific set of proteins are often analyzed by immunohistochemistry (Table 3). One or more antibodies have been used to characterize the proliferation and differentiation of the epidermis (basal, intermediate and superficial CKs). Moreover, a set of antibodies can be used to demonstrate if the newly-formed epidermis was or not repopulated by melanocytes (Melan A, S-100), Merkel (CD20) and/or Langerhans (CD1a) cells. The components of the cell–cell and dermoepidermal junction are evaluated through the identification of some specific markers. Cell–cell interactions can be demonstrated by immunohistochemistry against some desmosomal (e.g., desmoplakin, plakoglobin) or hemidesmosomal (α6β4 integrin) proteins. The basal membrane can be identified by histochemistry (PAS, silver stains) or immunohistochemistry (collagen IV, VII or laminin).

Regarding the degree and quality of the blood, lymphatic and nerve supply, these crucial elements can be assessed histologically. Blood vessels can be easily identified and classified by routine or histochemical methods. However, more specific analyses can be done by using endothelial (Von Willebrand Factor, CD31), basal membrane (laminin or collagen type IV) or SMA [14], which also allows to differentiate them from lymphatic vessels, positive for D2-40 protein [5]. Actually, it is also possible to differentiate vascular elements of the superficial and/or deep vascular plexus by using a-SMA and smoothelin [14]. With respect to the nerve supply, it is well-known that peripheral nerves can progressively reinnervate distal target organs, including the skin. Well-structured nerve fascicles can be identified with conventional histological techniques or myelin histochemical methods, as well as encapsulated nerve endings. However, free nerve endings or small regenerating fascicles are more difficult to identify. The Schwann cells can be identified by several markers, being the S-100 protein the most frequently used. In addition, neuronal axons are positive for neurofilaments, GAP-43 and PGP 9.5 proteins [135,139].

On the other hand, TEM has been used for a precise examination of highly specific epidermal and/or dermal features. As mentioned above, through this methodology it is possible to observe the ultrastructural details at the intracellular, intercellular and extracellular levels. TEM can be used to confirm, or as an advanced evaluation, of routine histology or histochemistry, but this will be of great scientific value if used to confirm highly specific immunohistochemical findings. In this context, TEM will allow a comprehensive characterization of the epidermal layers, the cell–cell or cell–ECM interactions. In the case of the basal membrane, the histochemistry or immunohistochemistry will provide information about if this structure was formed or not, but only TEM will provide pathognomonic images of its main elements (lamina basalis, lamina reticularis). Indeed, some authors used this technique for this specific propose [41].

Concerning the dermal ECM, histochemistry and immunohistochemistry will demonstrate the synthesis and organization of the main fibers and non-fibrillary ECM molecules while TEM will confirm the presence and organization of the collagen, reticular or elastic fibers [63,65,67,111]. In this regard, Lamme et al. used TEM to distinguish newly-synthesized collagen from the collagen-based scaffold used, and it was also a useful way to identify the myofibroblasts with their characteristic stress fibers [140]. Unfortunately, TEM only allows to evaluate very small samples losing the complete context of the whole healing process. For this reason, TEM is most suitable to answer specific research questions and it is less appropriate to take general conclusions about the whole wound healing process, where conventional histological techniques still represent a better option [86].

### 8.3. Molecular Biology

Histological study continues to be the most used and most informative diagnostic method. It provides global information and reports on some specific items. However, histological techniques are not able to quantify and evaluate some molecular and metabolic processes. In this context, molecular biology may proffer a valuable, fast and quantitative tool to evaluate the in vivo performance of tissue-engineered constructs. These procedures have increased the ability of analyze and describe basic molecular factors related to wound healing of skin [159].

Histology provides global information about the processes and some specific elements. However, the information provided may be lacking in terms of the quantification and evaluation of some molecular and metabolic processes and therefore molecular biology plays a fundamental role in the analysis and description of these processes. In fact, the histological examination is easy and accessible to perform. Molecular biology solves some of these drawbacks and generates a valuable, impartial and moderately early tool to evaluate the in vivo performance of tissue engineering constructs [159]. A practical tool in molecular analysis has been the Western blotting, a powerful technique used to detect specific protein molecules from among a combination of proteins, evaluate the size of a protein of interest and measure the amount of protein expression. It is based on an electrophoretically-separated sample using antibodies [208]. Western blot helps us to reaffirm the histological findings and makes a semiquantitative analysis of the expression of various markers, such as epithelial or dermal markers.

The proteomic analysis by Western blot will confirm all those results that have been obtained by a conventional histological examination and will contribute with a semiquantitative analysis of the expression of the different markers, among which the epithelial markers stand out.

PCR allows highly specific genes to be seen, which differentiates it from immunohistochemistry and the Western blot technique that recognize proteins. These genes can be related to differentiation, proliferation, signaling pathways, etc. Scherer et al. used quantitative RT-PCR to analyze the RNA expression levels in explanted mouse skin of the proangiogenic vasculature endothelial growth factor, the urokinase plasminogen activator receptor involved in cell migration, the inflammatory cytokine interleukin 1b, and matrix metalloproteinases 3 and 9 involved in extracellular matrix remodelling [157].

Finally, the gene expression microarrays technique allows us to determine the expression of many genes at a quantitative level. Klingenberg et al. used gene expression microarrays to evaluate changes in the gene expression profile of human cells in a cultured skin substitute after grafting onto mouse full-thickness wound [107]. Lammers et al. used this methodology to analyze changes in biological processes that occurred in a collagen-based acellular skin construct after implantation in a rat full-thickness wound model [159].

In conclusion, regardless of the great value that genetic analyzes provide, it is recommended that they be accompanied by histological analyses, generating an overall structural analysis.

### 8.4. Biomechanical Characterization

Mechanical properties of TESSs are largely related to its collagen-fiber architecture and kinematics and they are elementary for a correct functioning. It is remarkable that this aspect of artificial skin is often not assessed [209].

Yannas et al. determined the peeling force of their construct after transplantation, but the method used for this determination was not mentioned [210]. Pandit et al. tested the mechanical strength (ultimate tensile strength, stiffness, and failure strength) using an Instron tester to evaluate 1 × 4 cm skin strips in uniaxial tension and calculated the ultimate tensile strength, modulus of elasticity, and strain-to-failure [144]. Biomechanical analyses provide highly valuable information which could help to understand the histological and even molecular results. However, the high number of large samples needed may explain the limited use of biomechanical analyses in this field [86].

## 9. Conclusions

To conclude, diverse evaluation methods are available for the complete and comprehensive characterization of TESSs. Fortunately, there is more than one method available for the assessment of ex vivo and in vivo TESSs parameters. Therefore, it is important to choose those methods that permit a complete evaluation, in the most efficient and informative approach, of the TESSs generated (Figure 6).

Based on the studies included in this review, we can conclude that there is a large heterogeneity in the characterization of TESSs. Hence, it is necessary to improve the preclinical quality controls in this field defining some minimal criteria to ensure a complete, homogeneous and efficient preclinical characterization of these biomedical products to ensure their future clinical translation.

According to the information available, the most successful TESSs are those in which high levels of biomimicry and functionality are achieved. It is essential to confirm these features. In this review we found that histology represents one of the pillars of TESSs ex vivo and in vivo quality controls and for this reason we think that the tables presented in this review can be an invaluable help to know the quality requirements presented by the different constructs.

Therefore, for characterization of TESSs, we recommend a descriptive histological analysis with routine stainings or trichrome methods, and then selection of some key histochemical and immunohistochemical methods that efficiently demonstrate the level of epidermal and/or dermal biomimicry achieved.

In the case of the in vivo studies, the skin regeneration due to the use of TESSs should be demonstrated, but it is also important to evaluate the stromal remodeling, neovascularization and reinnervation processes, and, not least, the host immunological response. Even better and highly specific features can be demonstrated by the use of transmission electron microscopy, especially at the intracellular and extracellular level.

Despite the versatility offered by the histological analyses, there are always some limitations, and it is advisable to complement these results with molecular biology and functional tests. Molecular biology will provide highly valuable semiquantitative information about certain proteins or genes whereas functional tests will demonstrate the suitability of the TESSs generated for future in vivo preclinical studies. Finally, the TESSs should have adequate biomechanical properties to facilitate their implantation and suturing and avoid their contraction or rupture. In this sense, biomechanical characterization, an essential part in many TE applications, has been poorly studied in this field and thus there is also an important need for biomechanical evaluation of TESSs to approximate a successful clinical function.

In conclusion, it is evident that more standardized criteria for the characterization and validation of TESSs are necessary, especially when the aim of these models is clinical practice, where they are still urgently needed.

## Figures and Tables

**Figure 1 life-11-01033-f001:**
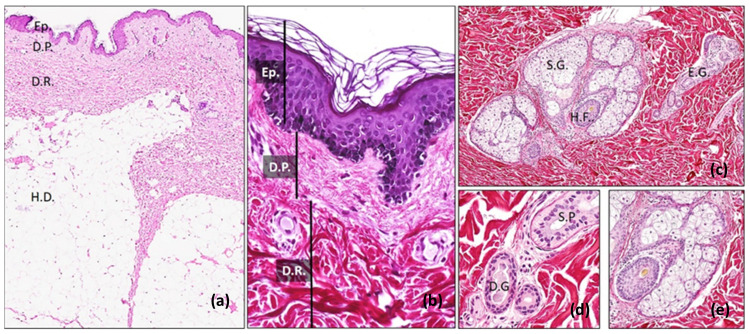
Human skin histology. In the low magnification histological section stained with haematoxylin-eosin (**a**) and higher magnification stained with Fontana–Masson picrosirius histochemical method (FMPS) (**b**) the three main layers of the skin, the epidermis (Ep.), papillary dermis (D.P.), reticular dermis (D.R.) and hypodermis (H.D.) are evident. In the images (**c**–**e**), stained with (FMPS), main skin appendages are shown. At low magnification it is possible to identify the sebaceous gland (S.G.), hair follicles (H.F.) and sweat glands or eccrine glandes (E.G.), the latter composed by a secretory portion (S.P.) and duct (D.G.). Original pictures (Department of Histology, University of Granada, Spain).

**Figure 2 life-11-01033-f002:**
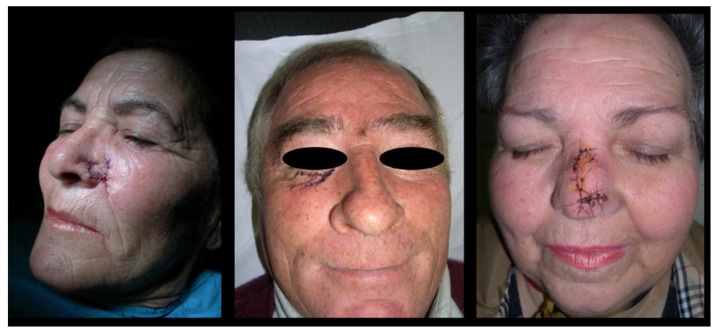
Clinical photos after local flap surgery on patients with different skin malignancies. Original pictures (Dermatology Unit, San Cecilio University Hospital, Granada).

**Figure 3 life-11-01033-f003:**
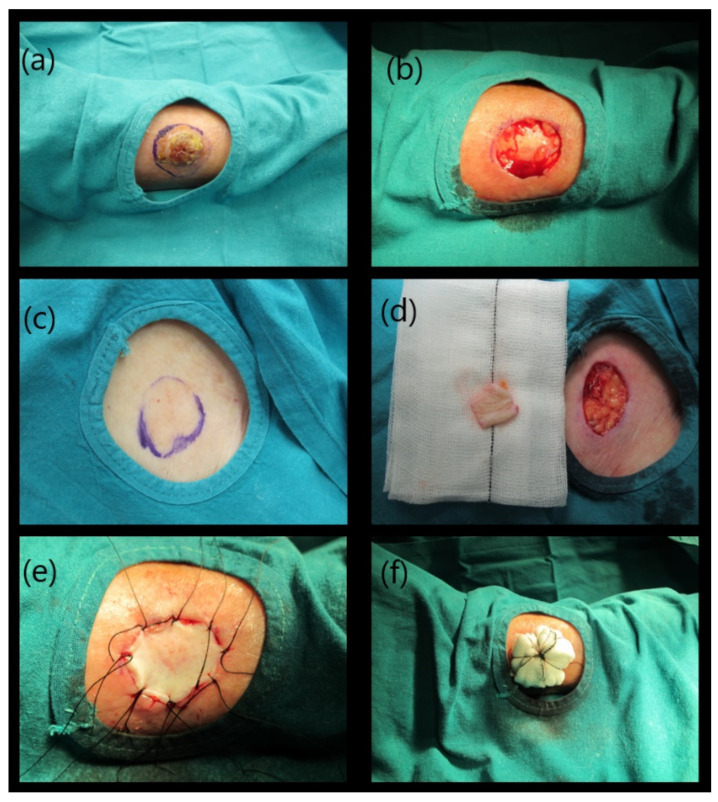
Reconstructive surgery based on the use of skin autograft. Images correspond to a patient who suffered a squamous cell carcinoma in her leg (**a**). Appearance of surgical defect after tumor resection (**b**). Donor skin obtained from the own patient (**c**,**d**). Reconstruction of skin defect using autograft (**e**,**f**). Original pictures (Dermatology Unit, San Cecilio University Hospital, Granada).

**Figure 4 life-11-01033-f004:**
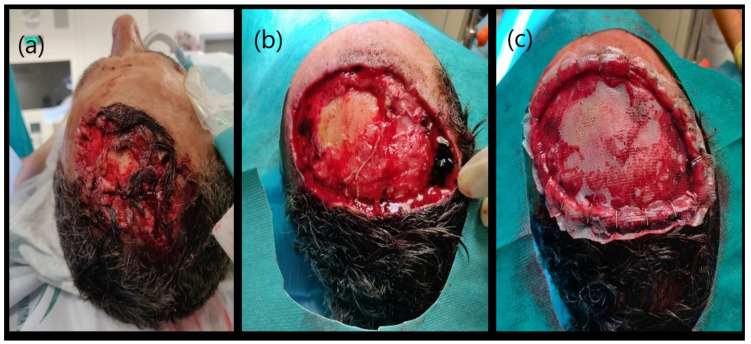
Surgical application of a biosynthetic skin substitute. This is a graphic example of the use of a Biobrane^®^ (Smith & Nephew, London, UK) on a patient after car accident. The wound before treatment (**a**), after the removal of necrotic tissue from the wound bed and surrounding damaged skin by surgical debridement (**b**) and the wound covered by the skin biosynthetic substitute (**c**). Original pictures (Dermatology Unit, San Cecilio University Hospital, Granada).

**Figure 5 life-11-01033-f005:**
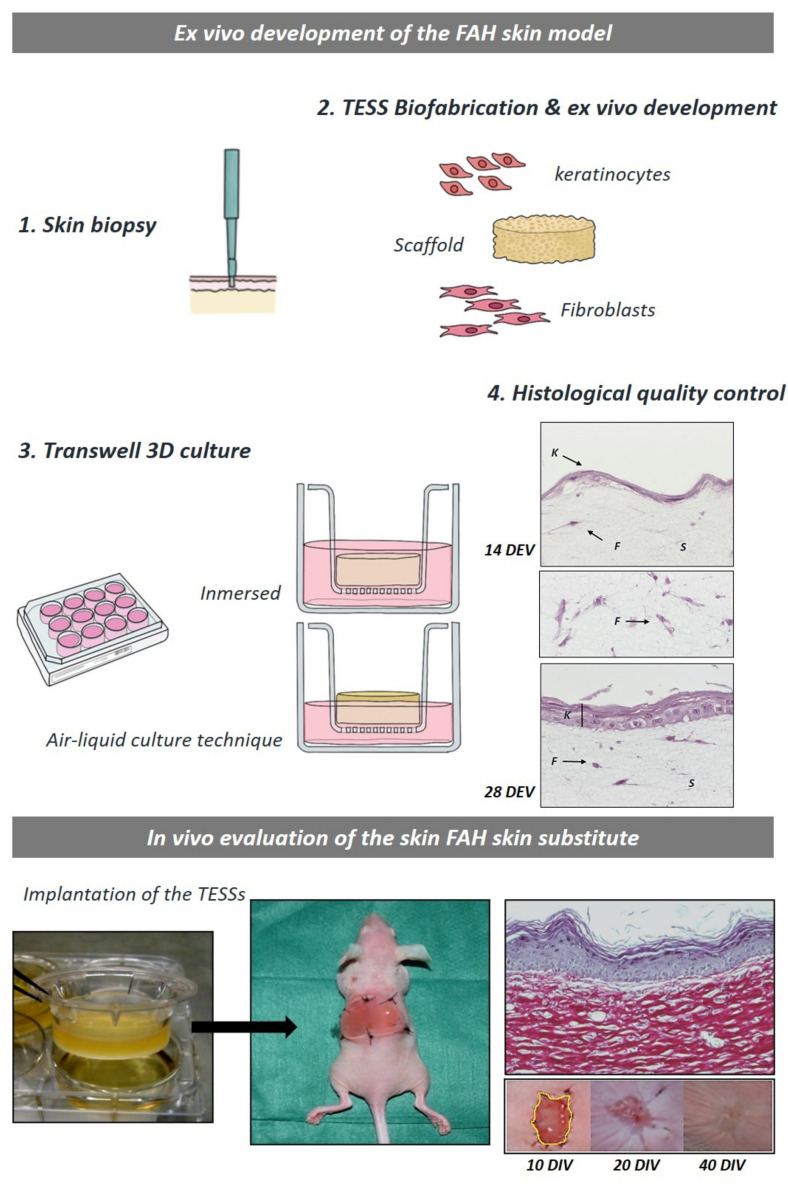
Schematic representation of the generation and in vivo evaluation of a human fibrin-agarose TESSs. The main steps for the generation of this skin model are represented from 1–3. The histological quality control of the epidermal and stromal development within the fibrin-agarose hydrogels are shown in step 4. The images of the in vivo evaluation show the macroscopic appearance of the skin substitute, the surgical implantation on nude mice, the macroscopic evolution of the wound healing process and the histological features of the dermo-epidermal junction. (FAH: Fibrin agarose hydrogels; DEV: devices; DIV Division).

**Figure 6 life-11-01033-f006:**
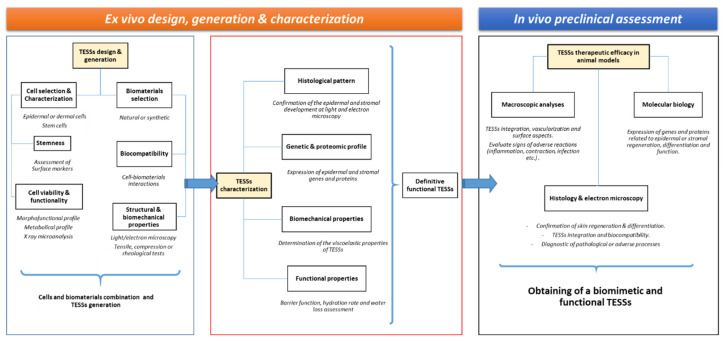
Summary of key aspects of the ex vivo and in vivo characterization methods often needed for the generation of TESSs.

**Table 1 life-11-01033-t001:** Strategies for skin repair: main advantages and limitations. Adapted from [16].

	Advantages	Limitations
**Autografts**	‘Gold standard’ in skin regeneration Good adhesion to the wound bed Provide pain relief No risk of rejection	Limited availability of donor sites Induce scar formation Patient morbidity Painful Increase risk of infection Lengthy hospital stays
**Allografts/Xenografts**	Temporary prevention of wound dehydration and contamination Promote angiogenesis Incorporate into deep wounds Alleviate the pain experienced by patients	Limited availability Lead to immune rejection Inflammation at the wound site Transmission of diseases
**Dressings**	Create and maintain a moist wound environment Can be made from a wide range of materials with different properties Ability to hydrate the wound and remove excess exudate	Low adhesion to the wound bed Inability to promote the regeneration of lost skin, in particular the dermal layer
**Tissue-engineered skin substitutes**	Promote the regeneration of dermis and epidermis Prevent fluid loss and provide protection from contamination May deliver extracellular matrix components, cytokines, growth factors and drugs to the wound bed Enhance the healing process Can be used in combination with autografts	High manufacturing costs Mechanical fragility Difficult handling Poor adhesion to the wound bed Possibility of immune rejection and transmission of diseases (allogeneic skin cells) Inability to promote the regeneration of full-thickness wounds Poor vascularization Impossibility of reproducing skin appendages

**Table 2 life-11-01033-t002:** Overview of studies describing the ex vivo evaluation methods.

Construct	Macroscopic Evaluation	Cell Viability	Histology	Immunohisto-Chemistry	Gene Expression	Electron Microscopy	Epidermal Barrier
Collagen-GAG-chitosan + FBs & KCs [91]			HE			X	
Collagen-GAG + human FBs & KCs [92]			HE				SEC
Fibrin + human FBs & KCs [62]			HE	Keratin 10, (pan)cytokeratin, laminin, type IV collagen			
Collagen-GAG + human FBs & KCs [93]			Toluidine blue				
Human DED + KCs vs RHE [94]		MTT	HE	Keratin 1, keratin 6, keratin 10, SKALP, transglutaminase I, involucrin, loricrin, SPRRs			
DED + collagen + human FBs & KCs [95]			HE	Keratin 1, involucrin, loricrin, filaggrin			
Collagen-GAG + human FBs & KCs +/− Vit C [96]		BrdU, MTT	HE	Collagen IV, collagen VII, laminin 5		X	SEC
Collagen-GAG + human FBs & KCs [97]	X		HE				
Collagen-GAG-chitosan + serum + human FBs & KCs [53]			Hematoxylin-phloxine-saffron	Keratin 10, keratin 14, transglutaminase, fibronectin, elastin, fibrillin 1, filaggrin, laminin, involucrin, integrin, collagen I, III, IV, V, Ki67		X	
Collagen + human FBs & KCs & melanocytes [98]			HE	Integrin, fibronectin, laminin, collagen IV, HLA-ABC			SEC
Collagen-GAG or PLGA +/− human FBs & KCs [99]	X	Live/dead	HE	(Pan)cytokeratin, laminin, collagen I, HLA-ABC		X	
Acellular human dermis + human FBs & KCs [61]			HE	a-SMA, collagen IV, VII, BP180 antigen (collagen XVII), Ki67			
Collagen-GAG + human KCs & FBs [100]		FdA, MTT	HE				
Gelatin-acrylamide + human BMSCs [101]		MTT		(Pan)cytokeratin, e-cadherin anti-CD13, CD34, CD44, CD45, CD49b, CD81, AC133, SH2, SH3		X	
PLLA vs PLGA + human KCs & FBs [63]		MTT	HE, Picrosirius red staining	Anti CD31, CD68, CD45RC		X	
Collagen-GAG + human FBs & KCs [102]		MTT	Toluidine Blue				Transepidermal water loss, H_2_O penetration, 14C-niacinamide permeability
Dermal component + human healthy vs psoriatic FBs & KCs [103]	X		Masson’s trichrome	Keratin 10, involucrin, loricrin, filaggrin, laminin V, Ki67			
Collagen-elastin + pancreatic SCs [70]		MTT, DAPI	HE	Keratin 10, keratin 14, fillagrin	PDX-1, GATA-1 genes		
Poly-N-acetyl-glucosamine vs cellulose [104]		Trypan blue, MTT				X	
DED + KCs & FBs [105]			HE	Keratin 5, keratin 10, integrin	Collagen, bFGF, TGFb1 mRNA	X	
Collagen + SGC + EGF + human FBs & KCs [106]		MTT	HE	Keratin 7, keratin 14, keratin 19, CEA			
Collagen-GAG + human FBs & KCs [107]			HE	KRT2, KRT15, loricrin, CILP, POSTN, OGN	DEFB4, KRT2, S100A7A, S100A12, SPRR2C, LOR, CD36, TCNI, GDA		
DED + collagen + human FBs & KCs [108]			HE				
FDM vs LEM vs FTM [109]			HE	Keratin 10, keratin 16, filaggrin, loricrin, involucrin, aquoporin 3		X	Benzocaine diffusion
Fibrin-agarose + human FBs & KCs [65]			HE, Picrosirius red staining, Alcian blue, Gomori, Orcein	Keratin 1, keratin 5, keratin 10, filaggrin, involucrin		X	
Human KCs [110]		MTT	HE				OCT
Fibrin/collagen + KCs +/− FBs, SVF, ASCs [111]			HE	Keratin 16, keratin 17, a-SMA, laminin 5, collagen I, anti CD31, CD34, CD73, CD90, aSMA, DAPI, vWF, Ki67		X	
Collagen + ADRCs [112]			HE	Anti CD31, CD45, CD90, CD34			
DED + hAECs + human FBs			HE, Periodic acid-Schiff	Keratin 10, keratin 14, keratin 18, keratin 19, filaggrin, laminin, desmoglein, collagen IV, Ki67		X	
Pegylated fibrin + ASCs [113]			HE, Alizarin Red S, Oil Red O, Sirius red/Fast green	a-SMA, anti CD68, CD206			
Collagen-GAG +/− ad-MVF [75]			HE, Sirius red	(Pan)cytokeratin, anti CD31, GFP/CD31			
S-dECM vs Collagen type I-HSE bioink [78]		Live/dead	HE, Masson’s trichrome, Alcian blue, laminin, DAPI	Anti CD31, CD34, CD133, CD45			Wettability, permeability, SEC
3D pigmented human skin construct [114]	X		HE, Fontana Masson	Keratin 1, keratin 6, collagen IV, VII, anti HMB45		X	
Type I collagen + gelatin-collagen microparticles and Aloe vera [115]		MTT	HE				
FN-G + HUVEC + FBs + KCs (87)	X	Live/dead	HE, Masson’s trichrome	Laminin 5, anti CD31			
Fibrin-agarose + MSCs [67]			HE, Picrosirius red, Periodic acid-Schiff, Alcian blue	Keratin 5, keratin 10, filaggrin, HLA I-II	X	X	
PCL-NCs/Cur + EnSCs [116]		MTT					Wettability
Lesional psoriatic skin [117]			HE	Keratin 16, anti CD3, CD23, CD28	IL17, IL8		

Haematoxylin-eosin staining (HE); surface electrical capacitance (SEC); glycosaminoglycan (GAG); fibroblasts (FBs); keratinocytes (KCs); 3- (4,5-dimethylthiazol-2-yl)-2,5-diphenyltetrazolium bromide (MTT); antibodies to skin-derived antileukoproteinase (SKALP); small proline rich proteins (SPRRs); de-epidermized dermis (DED); 5-bromo-2¢-deoxyuridine (BrdU); reconstructed human epidermis (RHE); vitamine (Vit); a-smooth muscle actin (a-SMA); bone-marrow-derived mesenchymal stem cells (BM-MSCs); fluorescein diacetate (FdA); poly-L-lactide (PLLA); three poly (D,L)-lactide-co-glycolide (PLGA); SCs (stem cells); cultured sweat gland cells (SGC); epidermal growth factor (EGF); 4’,6-diamidino-2-phenylindole (DAPI); transforming growth factor beta (TGFb); b-defensin 2 (DEFB4 gene), the differentiation-specific keratin 2 gene (KRT2), osteoglycin (OGN), cartilage intermediate layer protein (CILP), periostin (POSTN), S100 calcium-binding protein A7A (S100A7A), S100 calcium-binding protein A12 (S100A12), loricrin (LOR), small proline-rich protein 2C (SPRR2C), transcobalamin I (TCN1), guanine deaminase (GDA), fibroblast-derived matrix model (FDM); Leiden epidermal model (LEM); full-thickness collagen model (FTM); adipose stromal vascular fraction (SVF); adipose stromal cells (ASCs); Von Willebrand factor (vWF); adipose derived regenerative cells (ADRCs); human amniotic epithelial cells (hAECs); adipose tissue-derived microvascular fragments (ad-MVF); skin-derived extracellular matrix (S-dECM); fibronectin and gelatin matrix (FN-G); human umbilical vein endothelial cells (HUVEC); mesenchymal stem cells (MSCs); electrospun polycaperlactone (PCL); curcumin-loaded chitosan nanoparticle (NCs/Cur); human endometrial stem cells (EnSCs). X: Examination performed in the study.

**Table 3 life-11-01033-t003:** Overview of studies describing the in vivo evaluation methods.

Construct	Macroscopic Evaluation	Cell Viability	Histology	Immunohisto Chemistry	Gene Expression	Electron Microscopy	Epidermal Barrier
Polivinyl alcohol [138]	X		HE				
Collagen-GAG +/− KCs [139]	X		HE				
Collagen + rat FBs [52]	X		HE				
Collagen, polyglactin or PEU [78]	X		HE	Laminin			
Polyglactin + human FBs & KCs [116]	X		HE	Involucrin, laminin			
Collagen + rat KCs +/− fibrin [41]	X			Collagen IV		X	
Collagen-GAG-chitosan [59]	X		HE				
Acellular human DED + human KCs [31]	X		HE, Masson’s trichrome	Keratin 10, keratin 16, involucrin, laminin, collagen IV			
Collagen-GAG + human FBs & KCs [110]	X		HE				SEC
Collagen-elastin hydrosylate [140]			HE	Fibronectin, laminin, chondroitin sulfate, elastin, vWF		X	
Collagen -GAG + porcine KCs [141]	X		HE				
Fibrin + human FBs & KCs [44]			HE	Laminin, type IV collagen, (pan)cytokeratin, keratin 10			
Collagen-GAG + porcine KCs [142]	X		HE	(Pan)cytokeratin, integrin, involucrin, laminin, collagen VII, factor VIII, Ki67			
Collagen-elastin hydrosylate + porcine FBs [143]			HE, Herovici stain	Vimentin, vWF			
Collagen-GAG + human FBs & KCs [93]	X			HLA-ABC			SEC
Collagen +/− TGFB [144]			HE, Modified Masson’s trichrome				
Collagen-elastin hidrosylate +/− porcine FBs [145]	X		HE	Elastin, a-SMA			
Collagen-GAG + human FBs & KCs [97]	X		HE				
Collagen-GAG + human FBs & KCs +/− Vit C [96]	X	BrdU, MTT	HE	Collagen IV, collagen VII, laminin 5		X	SEC
Collagen-GAG + fibrin & porcine KCs [146]	X			Keratin 6, keratin 14			
Gelatin-B-glucan +/− human FBs & KCs [60]			HE				
PGA +/− murine FBs +/− transfected PDGF [147]			HE	Anti-PDGF	PDGF-B		
Biological and synthetic scaffolds +/− porcine FBs [148]	X		HE, Masson’s trichrome				
Plasma + human FBs & KCs [149]	X		HE, Masson’s trichrome	(Pan)cytokeratin, keratin 5, keratin 10, involucrin, laminin, loricrin, vimentin			
GAG −/− porcine FBs +/− porcine KCs [150]	X		HE	Keratin 6, collagen VII	Autosomal DNA, male DNA		
Collagen +/− GAG or PEGT/PBT [151]	X		HE, Sirius red	vWF, vimentin, Ki67			
Collagen-GAG or PLGA +/− human FBs & KCs [99]	X		HE	(Pan)cytokeratin, laminin, collagen I, HLA-ABC		X	
Acellular human dermis + human FBs & KCs [61]	X						
Collagen + human FBs & KCs & melanocytes [98]	X		HE	HLA-ABC			
PEGylated-RGD gelatin & KGF-1 [82]	X		HE				
GAG + porcine KCs [152]	X		HE, Mallory’s trichromate				
Human DED [153]	X		HE, elastica von Giesson	a-SMA			
Fibrin +/− eNOS expressing vector [154]	X		HE, Masson’s trichrome, Picrosirius red	Anti CD31, e-NOS			
Gelatin-acrylamide + human BMSCs [101]	X		HE	(Pan)cytokeratin, e-cadherin anti-CD13, CD105			
Agar-collagen [155]	X		HE				
Collagen-GAG + human KCs & FBs [100]		FdA, MTT	HE				
PLLA vs PLGA + human KCs & FBs [63]	X		HE	Anti CD31, CD68, CD45		X	
Collagen + human FBs & KCs [156]	X		HE, orcein, periodic acid-Schiff	(Pan)cytokeratin, vimentin, HLA-DR, HBG			
Silk fibroin-chitosan + ASCs [68]	X		HE	Keratin 19, a-SMA, vWF, Ki67			
Collagen-elastin + pancreatic SCs [70]	X	MTT, DAPI	HE	Keratin 10, keratin 14, fillagrin	PDX-1, GATA-1 genes		
Poly-N-acetyl-glucosamine vs cellulose [157]	X		HE	PECAM-1, anti CD45, CD31, Ki67, p63	MMP3, uPAR, VEGF		
Collagen-GAG + human FBs & KCs [107]	X		HE	KRT2, KRT15, loricrin, CILP, POSTN, OGN	DEFB4, KRT2, S100A7A, S100A12, SPRR2C, LOR, CD36, TCNI, GDA		
Hyalluronic acid-collagen +/− human FBs [158]	X	MTT	HE	Vimentin			
Collagen-GAG + GFs [159]			HE	Muscle-specific desmin, anti-CD8 alpha, collagen IV	X		
Collagen + SGC + EGF + human FBs & KCs [160]	X		HE				
Collagen-GAG + GFs [161]	X		HE, Elastin von Gieson, Verhoeff’s elastic tissue, Masson’s trichrome	a-SMA, elastin, collagen I, III, IV, dermatan sulfate		X	
Fibrin-agarose + human FBs & KCs [65]			HE, Picrosirius red staining, Alcian blue, Gomori, Orcein	Keratin 1, keratin 5, keratin 10, filaggrin, involucrin		X	
Collagen + BM-MSC +/− EGF [69]	X		HE	Keratin 5, CEA			
Type I collagen gel + collagen-elastin [162]	X		HE, Masson’s trichrome	a-SMA			
DED + Collagen-GAG vs Collagen-elastin + KCs [163]	X		HE, elastica van Gieson	Keratin 10, filaggrin, cathepsin V, loricrin			
Fibrin/collagen + KCs +/− FBs, SVF, ASCs [111]	X		HE	Keratin 16, keratin 17, a-SMA, laminin 5, collagen I, anti CD31, CD34, CD73, CD90, aSMA, DAPI, vWF, Ki67		X	
Collagen + ADRCs [112]	X		HE, Masson Trichrome	a-SMA, anti CD31, CD45, CD90, CD146			
Pegylated fibrin + ASCs [113]	X		HE, DAPI, Alizarin Red S, Oil Red O	a-SMA, lectin, anti CD68, CD206			
3D printing vHSEs [76]	X		HE	Keratin 10, keratin 14, loricrin, anti CD31, Ki67			
Collagn-GAG +/− ad-MVF [75]			HE	(Pan)cytokeratin, anti CD31, GFP/CD31			
S-dECM bioink +/− EPCs + ASCs [78]			HE	Keratin 10, anti CD31			
FN-G + HVEC + FBs + KCs [128]	X		HE, Masson’s trichrome	Anti CD31, HLA-ABC			
Type I collagen + gelatin-collagen microparticles and Aloe vera [115]	X	MTT	HE				
Fibrin-agarose + MSCs [67]			HE, Picrosirius red, Periodic acid-Schiff, Alcian blue	Keratin 5, keratin 10, filaggrin, HLA I-II	X	X	
PCL-NCs/Cur + EnSCs [116]	X		HE, Masson’s trichrome				Wettability

Glycosaminoglycan (GAG); fibroblasts (FBs); keratinocytes (KCs); PEU: polyether urethane; de-epidermized dermis (DED); transforming growth factor beta (TGF-β); polyglycolic acid (PGA); platelet-derived growth factor (PDGF); vascular endothelial growth factor (VEGF); polyethylene glycol (PEGT); polybutylene terephthalate (PBT); three poly (D,L)-lactide-co-glycolide (PLGA); arginine-glycine-aspartic acid (RGD); endothelial nitric oxide synthase (eNOS); b-defensin 2 (DEFB4 gene), the differentiation-specific keratin 2 gene (KRT2), osteoglycin (OGN), cartilage intermediate layer protein (CILP), periostin (POSTN), S100 calcium-binding protein A7A (S100A7A), S100 calcium-binding protein A12 (S100A12), loricrin (LOR), small proline-rich protein 2C (SPRR2C), transcobalamin I (TCN1), guanine deaminase (GDA); bone-marrow-derived mesenchymal stem cells (BM-MSCs); vitamine (Vit); poly-L-lactide (PLLA); adipose Stromal Cells (ASCs); SCs (stem cells); cultured sweat gland cells (SGC); epidermal growth factor (EGF); growth factors (GFs); adipose Stromal vascular fraction (SVF); Von Willebrand factor (vWF); adipose-derived regenerative cells (ADRCs); vascularized human skin equivalents (vHSEs); adipose tissue-derived microvascular fragments (ad-MVF); skin-derived extracellular matrix (S-dECM); fibronectin and gelatin matrix (FN-G); human umbilical vein endothelial cells (HUVEC); curcumin-loaded chitosan nanoparticle (NCs/Cur); human endometrial stem cells (EnSCs); platelet endothelial cell adhesion molecule 1 (PECAM-1), matrix metalloproteinases (MMPs); messenger RNA levels related to migration (uPAR); blood-group antigen (HBG). X: Examination performed in the study.

**Table 4 life-11-01033-t004:** Overview of immunohistochemical staining most used to evaluate tissue-engineered skin constructs. Basement membrane (BM); blood vessels (BV).

	Technique	Tissue/Cells	Reference
**Epidermal**	P63	Migrating keratinocytes	[162]
	(Pan)cytokeratin	Keratinocytes	[68,75,77,92,101,107,117,165,166,167]
	Keratin 6	Hyperproliferative keratinocytes	[94,107,146,168]
	Keratin 5	Basal keratinocytes	[112,128,149,169]
	Keratin 14	Basal keratinocytes	[76,89,90,115,117,146,160]
	Keratin 15	Basal keratinocytes	[107]
	Keratin 16	Basal keratinocytes	[111,128,168]
	Keratin 19	Basal keratinocytes	[68,89,90,138,160]
	Keratin 1	Suprabasal keratinocytes	[53,94,95,117,128]
	Keratin 10	Suprabasal keratinocytes	[16,76,78,90,94,107,115,128,149,168,169]
	CD185	Keratinocyte stem cell markers	[94]
	Involucrin	Cornified envelope keratinocytes	[36,46,67,78,95,103,117,128,149,168,170]
	Loricrin	Cornified envelope keratinocytes	[76,94,95,103,107,114,149,168]
	Fillaggrin	Granular keratinocytes	[53,67,90,95,103,114,115,128,168,169]
	Transglutaminase	Granular keratinocytes	[53,89]
	Integrin	Attaching keratinocytes	[98,105,117,147,171,172]
	6-Integrin	Keratinocyte stem cell markers	[53,94]
**Dermal**	A-smooth muscle actin	Myofibroblasts & mature blood vessels	[61,68,82,107,111,112,117,147,161,173]
	Type I collagen	(Newly formed) dermis	[53,89,92,111,161,173,174]
	Type III collagen	(Newly formed) dermis	[53,89,161,174]
	Type V collagen	(Newly formed) dermis	[53,98,169,174]
	Elastin	Elastic fibers	[36,46,107,161]
	Fibrillin-1	Microfibrils (elastic fiber formation)	[53,174]
	Fibronectin	Wounded dermis	[53,89,98,107,174]
	Vimentin	Fibroblasts	[68,117,143,149,151,173,175,176]
	CD68	Monocytes/macrophagues	[63,107]
	CD45RC	Lymphocytes	[63,162]
**BM**	Type VII collagen	Basement membrane	[53,89,107,117,143]
	Desmoglein	Basement membrane	[90]
**BM and BV**	Type IV collagen	Blood vessels & basement membrane	[10,46,89,96,107,161]
	Laminin	Blood vessels & basement membrane	[10,53,78,89,92,96,98,107,114,128,149,162,170]
**BV**	CD31	Blood vessels	[63,75,78,112,114,128,154]
	Factor VIII	Blood vessels/endothelial cells	[117]
	PECAM-1	Blood vessels	[162]
	Von Willebrand factor	Blood vessels	[68,107,143,151,173]
	D2-40 protein	Lymphatic vessels	[5]
**Other**	Desmin	Muscle cells	[76]
	Ki67	Proliferating cells	[10,61,68,103,114,117,151,162]
	HMB-45	Anti-melanoma antibody	[107]
	S-100 protein	Schwann cells	[135]
**Labels**	Endothelial nitric oxide synthase	In situ transfected cells	[154]
	Vascular endothelial growth factor	In situ transfected cells	[76,177]
	Human leukocyte antigen (HLA)	Transplanted cell “label”	[92,117,143,169,173]
**Immune Response**	CD68	Macrophages	[63,106]
	CD45	Lymphocytes	[63]

**Table 5 life-11-01033-t005:** Biomechanical testing overview for the characterization of TESSs generated.

Reference	TESSs	Biomechanical Evaluation In Vitro	Biomechanical Evaluation In Vivo	Parameters
Kim et al. [78]	S-dECM bioink	Rheological properties		
Zahiri et al. [116]	PCL vs PCL/Gela vs PCL/Gela/NCs/Cur (+EnSCs)	Uniaxial loaded by tensile test		Tensile strength (MPa)
Freytes et al. [187]	Five different ECM scaffolds before and after treatment with peracetic acid (PAA): SS, SIS, UBS, UBM, UBS + UBM. This study also compared the mechanical properties of two- and four-layer ECM scaffolds	Tensile test Ball-burst test		Ball-burst strength (N)
Badylak et al. [188]	SISHRD	Ball-burst test	Ball-burst test	Survival time/Ball burst load (pounds)
Ko et al. [189]	SIS	Tensile strength test	Tensile strength test	Mean Tensile Strength (N/cm)
Gloeckner et al. [190]	Graftpatch^®^ (Clarivate, London, UK) vs SIS vs GLBP	Stress-based biaxial test		
Pandit et al. [144]	Collagen va collagen + TGF-B		Instron tester in uniaxial tension	Ultimate tensile strength, Stiffness, Failure strain
Shah et al. [191]	Decellularized human dermis	OCT and vibrational analysis	OCT and vibrational analysis	Resonant Frequency
Heraud et al. [192]	SFSE	Suction experimental device Cutometer 580^®^ (Microcaya, Bilbao, Spain)		Ue, the immediate elastic elongation; Uf, the total elongation; Uv, the viscous (creep) elongation; Ur, the immediate recovery; Ur/Ue, the elastic ratio; Ur/Uf, the relative elastic recovery; and Uv/Ue, the viscous ratio
Lafrance et al. [193]	hKCs seeded on our anchorage based a human type supplemented with elastin DE, I+III CG + GAGS	Indentation method		Deflection (A in mm)
Lafrance et al. [194]	FBs + type I bovine collagen	Tensile tests: indentation test		Tensile Strength
Zahouani et al. [195]	FBs + Human skin vs Dermal substitute (bovine collagens 95% type I, 5% type III + chitosan + chrondroitin-4, sulfate		New bio-tribometer working at a low contact pressure	Young modulus
Ahlfors et al. [196]	Collagen gel vs Fibrin gel vs CDM + vs native skin	Custom tissue inflation device		Failure tension, failure strain, and ultimate tensile strength (kPa)
Berthod et al. [197]	Collagen + chondroitins 4-, 6-sulphate + chitosan	Tensile strength test		Tensile Strength, ultimate elongation and Young’s modulus

Skin-derived extracellular matrix (S-dECM); electrospun polycaperlactone (PCL); gelatin (Gela); curcumin-loaded chitosan nanoparticle (NCs/Cur); human endometrial stem cells (EnSCs); human fibroblasts (hFBs); human keratinocytes (hKCs); collagen–glycosaminoglycan (C-GAG); canine stomach submucosa (SS); porcine small intestinal submucosa (SIS); porcine urinary bladder submucosa (UBS); porcine urinary bladder matrix (UBM); multilaminate small intestinal submucosa hernia repair device (SISHRD); glutaraldehyde-treated bovine pericardium (GLBP); growth factor beta (TGF-b); scaffold-free skin equivalent (SFSE); dermal equivalent (DE); collagen gel (CG); cell-derived matrix (CDM).

## Data Availability

Not Applicable.

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
