# Peer review of "Basic Quality Controls Used in Skin Tissue Engineering"

_life, 2021, doi:10.3390/life11101033_

Round 1
Reviewer 1 Report
In this manuscript, authors have extensively reviewed TESSs and factors of quality controls. It will certainly help skin tissue engineers for advanced research. However, there are several points that authors have to consider to advance their manuscript. Therefore, we have commented points that need additional revision. Furthermore, English language correction would improve the readability. Below are the comments for current manuscript. To improve the manuscript, please add or modify the statements by referring below comments. 1. Even though authors have used references about 3D bioprinted skin grafts, there are no information in the manuscript. Not only 3D bioprinting technology, there are multiple tissue engineering methods for skin tissue graft development. Please organize the skin tissue engineering methods (table) including 3D bioprinting methods. 2. Please add figure captions and legends to avoid confusion at Figure 1. 3. On Table 1. authors have listed the pros and cons about each method. However, less information are stated at manuscript. Please rephrase statements for better readability to readers. 4. If authors can correlate the regeneration mechanism and TESSs (quality controls), it will possess much more impact. 5. About Figure 2 to 4, did authors obtained the patient photographic authorization? 6. References are separated in some tables and not separated in others. Please separate the Reference and rearrange it in the table.Author Response
Reviewer 1
In this manuscript, authors have extensively reviewed TESSs and factors of quality controls. It will certainly help skin tissue engineers for advanced research. However, there are several points that authors have to consider to advance their manuscript. Therefore, we have commented points that need additional revision. Furthermore, English language correction would improve the readability. Below are the comments for current manuscript. To improve the manuscript, please add or modify the statements by referring below comments.
R/ We thank the reviewer for his kind and objective criticism of our manuscript. We proceed to respond all suggestions to improve the quality of this manuscript.
- Even though authors have used references about 3D bioprinted skin grafts, there are no information in the manuscript. Not only 3D bioprinting technology, there are multiple tissue engineering methods for skin tissue graft development. Please organize the skin tissue engineering methods (table) including 3D bioprinting methods.
R/. Dear reviewer, thank you very much for your valuable suggestion. We included information about the 3D printing technology in the “Skin tissue engineering” section of the revised manuscript as suggested (page x, lines xx).
- Please add figure captions and legends to avoid confusion at Figure 1.
R/ Dear reviewer, apologies for the confusion. The legend of the figure 1 was rewrite as suggested in order to avoid any kind of confusion.
- On Table 1. authors have listed the pros and cons about each method. However, less information are stated at manuscript. Please rephrase statements for better readability to readers.
R/. Dear reviewer, thank you very much for this valuable suggestion that really improve our manuscript. New information concerning these surgical techniques was included within the revised manuscript as suggested (Page X, lines X).
- If authors can correlate the regeneration mechanism and TESSs (quality controls), it will possess much more impact.
R/. We agree with this suggestion. The information is in the manuscript and based on your suggestion and we have included some sentences to connect or correlate the different sections of the manuscript.
- About Figure 2 to 4, did authors obtained the patient photographic authorization?
Author reply: Dear reviewer, thank you very much for this question. There is written informed consent for all those patients whose image appears in the manuscript and this information was included within the Informed Consent Statement of the revised manuscript.
- References are separated in some tables and not separated in others. Please separate the Reference and rearrange it in the table.
R/. Dear reviewer, we understand your observation, however, the format used is based on the aspects that we decided to highlight in each table. For this reason we prefer to keep the current format.
Reviewer 2 Report
The review paper presents a comprehensive overview on the methods for the characterization of tissue-engineered skin substitutes. The manuscript covers an interesting and timely topic, and is well-presented. Please see my comments/suggestions bellow.
1. A broad range of biomaterials have been designed to be used as more “chemically defined” skin ECM-mimics to create skin in vitro. Authors should include more examples of these biomaterials in tables 2 and 3 to cover a wide range of biomaterials and to provide the readers with a representative overview of biomaterials used to create tissue-engineered skin. Some examples of recent works include: 10.1088/1758-5090/aba503; 10.1016/j.biomaterials.2020.120287; 10.1039/C8MH00525G; 10.1089/ten.tea.2019.0319; 10.1002/adhm.201801019; 10.1016/j.actbio.2017.11.016; 10.1088/1758-5090/9/1/015006; 10.1088/1758-5090/aa71c8.
2. A key issue in the fabrication of skin relies on the selection of cells and their impact on the biological function of the constructs. In a review paper focused on the characterization of skin constructs and, therefore, their biological function, I would expect to find a short section or at least some sentences discussing this issue.
3. It would be more compelling if the authors include one figure summarizing key characterization methods that have been performed for the evaluation of skin constructs as well as key aspects that have been overlooked and should be considered in the future. Although this info is referred in different sections throughout the manuscript, it would be more valuable if one could find the information in an easier and condensed way.
Minor issues:
Line 32: references are missing in the sentence “Some of these engineered skin models showed promising ex vivo, in vivo and even clinical results”.
The presentation of Table 1 should be improved to allow easier distinction of the attributes between each strategy. In the current form it is hard to follow.
Line 515: “Safranina O” should be “Safranin O”.
Author Response
Reviewer 2
The review paper presents a comprehensive overview on the methods for the characterization of tissue-engineered skin substitutes. The manuscript covers an interesting and timely topic, and is well-presented. Please see my comments/suggestions bellow.
R/ Dear reviewer, thank you very much for your positive comments and scientific advice about our manuscript.
1.A broad range of biomaterials have been designed to be used as more “chemically defined” skin ECM-mimics to create skin in vitro. Authors should include more examples of these biomaterials in tables 2 and 3 to cover a wide range of biomaterials and to provide the readers with a representative overview of biomaterials used to create tissue-engineered skin.
Some examples of recent works include: 10.1088/1758-5090/aba503; 10.1016/j.biomaterials.2020.120287; 10.1039/C8MH00525G; 10.1089/ten.tea.2019.0319; 10.1002/adhm.201801019; 10.1016/j.actbio.2017.11.016; 10.1088/1758-5090/9/1/015006; 10.1088/1758-5090/aa71c8.
R/ Dear reviewer, your observation is correct, there is a wide range of biomaterials and biofabrication techniques in use in skin tissue engineering. In this sense, most of these references and well as other suggested by other reviewers were included within the “Skin tissue engineering” section in the revised manuscript. The following references were included:
DOI: 10.1089/ten.TEA.2019.0319
DOI: 10.1088/1758-5090/aba503
DOI: 10.1016/j.biomaterials.2020.120287
DOI: 10.1088/1758-5090/aa71c8
- A key issue in the fabrication of skin relies on the selection of cells and their impact on the biological function of the constructs. In a review paper focused on the characterization of skin constructs and, therefore, their biological function, I would expect to find a short section or at least some sentences discussing this issue.
R/ Dear reviewer, your advice is right. Stem cells represents a promising alternative for a wide range of applications, including skin tissue engineering. However, in the case of skin TE, most groups, and specially those focused on the generation of skin equivalent for clinical use, employ keratinocytes and fibroblasts, as these cells are the most efficient alternative in the field. However, and based on your suggestion, new information concerning the use of stem cells was included in the revised manuscript and in the corresponding tables.
- It would be more compelling if the authors include one figure summarizing key characterization methods that have been performed for the evaluation of skin constructs as well as key aspects that have been overlooked and should be considered in the future. Although this info is referred in different sections throughout the manuscript, it would be more valuable if one could find the information in an easier and condensed way.
R/. Dear reviewer, we appreciate your valuable advice. In this sense, we made a new figure (Figure 6) in which a summary of key aspects and quality control methods are shown for ex vivo and in vivo characterization of TESSs.
Minor issues:
The presentation of Table 1 should be improved to allow easier distinction of the attributes between each strategy. In the current form it is hard to follow.
R/ Dear reviewer, the Table 1 was modified accordingly and also based on the suggestions made by other reviewers.
Line 515: “Safranina O” should be “Safranin O”.
Author reply: It has been changed properly.
Reviewer 3 Report
In this study, the authors did a comprehensive article review with the title "Ex vivo and In vivo Quality Controls in Skin Tissue Engineering". Authors have described skin regeneration and treatment options (surgery, skin tissue engineering) from basic skin biology (structure, cells, expressing biomarker) to skin regeneration and treatment options (surgery, skin tissue engineering). In addition to this, the quality control parameters in the ex vivo and in vivo study set-up in TESS are summarized in a table. Although the authors' review topics are timely and interesting, they have critical limitations on integrity and quality for the following reasons:
- It appears that the authors did not conduct an appropriate plagiarism test. In the Ithenticate similarity test, this review paper has a similarity of 23%. This similarity starts from Abstract. Especially contents from two articles [1. Novel Approaches To Regenerative Medicine Of The skin and 2. Evaluation methods as quality control in the generation of decellularized peripheral nerve allografts] are used without paraphrasing. Authors should paraphrase and rewrite the article in their own style rather than copy and paste the article they are reviewing.
- The title of the paper is Ex vivo, In vivo quality control in skin tissue engineering, but the part about quality control in the content is not outstanding except for the table. Authors need to visualize what they want to emphasize through figures.
- Basic histology of the skin on Page 2 cannot cover all of the described content. "Biology of the Skin" seems more appropriate.
- In Regeneration of the skin, on Page 4, Line 143, the authors described, "The most common... are thermal injuries." How about trauma instead of thermal injury? The authors need to add a proper reference.
- Figures 2, 3, and 4 of Page 5 are not properly explained in the Manuscript. And it is questionable whether each figure is appropriate for the direction of this paper.
- The Manuscript requires extensive formatting.
- Examples of formatting errors throughout the Manuscript.
1) Page 7, Line 211-212 lined sentence.
2) Page 7, Dot in Table 1.
3) It is difficult to identify each item on Page 7 and Table 1 (Need separation line).
4) Page 9, Line 282-296, green-colored paragraph.
5) Duplicate of Acronyms.
Author Response
Reviewer 3
In this study, the authors did a comprehensive article review with the title "Ex vivo and In vivo Quality Controls in Skin Tissue Engineering". Authors have described skin regeneration and treatment options (surgery, skin tissue engineering) from basic skin biology (structure, cells, expressing biomarker) to skin regeneration and treatment options (surgery, skin tissue engineering). In addition to this, the quality control parameters in the ex vivo and in vivo study set-up in TESS are summarized in a table. Although the authors' review topics are timely and interesting, they have critical limitations on integrity and quality for the following reasons:
R/ We thank the reviewer for his kind criticism of our manuscript and we proceed to respond to suggestions to improve it.
- It appears that the authors did not conduct an appropriate plagiarism test. In the Ithenticate similarity test, this review paper has a similarity of 23%. This similarity starts from Abstract. Especially contents from two articles [1. Novel Approaches To Regenerative Medicine Of The skin and 2. Evaluation methods as quality control in the generation of decellularized peripheral nerve allografts] are used without paraphrasing. Authors should paraphrase and rewrite the article in their own style rather than copy and paste the article they are reviewing.
R/ Dear reviewer, the similarity has been reduced under 20% as it has been required in the rest of scientific journals
- The title of the paper is Ex vivo, In vivo quality control in skin tissue engineering, but the part about quality control in the content is not outstanding except for the table. Authors need to visualize what they want to emphasize through figures.
R/. Thank you for your observation. Our aim was to review the main quality controls used in skin tissue engineering. As our manuscript does not cover all the aspect as expected and based on your valuable information we decided to modify the title as follow: Basic quality controls used in skin tissue engineering.
- Basic histology of the skin on Page 2 cannot cover all of the described content. "Biology of the Skin" seems more appropriate.
R/ Thank you very much, it is a good suggestion, the section title was modified accordingly.
In Regeneration of the skin, on Page 4, Line 143, the authors described, "The most common... are thermal injuries." How about trauma instead of thermal injury? The authors need to add a proper reference.
R/. This is a valuable suggestion. The incidence of the injuries varied among countries, gender, age, etc. We agreed with your observation and thus this information was changed as suggested.
- Figures 2, 3, and 4 of Page 5 are not properly explained in the Manuscript. And it is questionable whether each figure is appropriate for the direction of this paper.
R/. Dear reviewer, more information concerning the surgical techniques illustrated in the images was added within the “current surgical strategies for skin repair” section, improving the description of these graphical content as suggested.
- The Manuscript requires extensive formatting.
R/. We have corrected all the suggested and new formatting errors that we have found
- Examples of formatting errors throughout the Manuscript.
1) Page 7, Line 211-212 lined sentence.
2) Page 7, Dot in Table 1.
3) It is difficult to identify each item on Page 7 and Table 1 (Need separation line).
4) Page 9, Line 282-296, green-colored paragraph.
5) Duplicate of Acronyms.
Round 2
Reviewer 3 Report
I appreciate the effort of the authors. However, even though total plagiarism was reduced significantly (less than 20% [in my ithenticate; 18%]), still lots of sentences are exactly the same as the previously published article.
Please check below
- Line 154-168; 10.3233/JCB-179004 and 10.1016/j.matbio.2015.08.001
- Table 1: https://www.medscape.com/viewarticle/782469_2
- Line 299-308; 10.1016/j.biomaterials.2008.03.037
- Line 638-643
- Line 652-658
- Line 701-712
- Line 850-856
- Line 867-878
- Line 885-892
Author Response
Thank you very much for your kind review.
We have corrected all the segments of the manuscript where there could be signs of plagiarism and we have written them in our own words to avoid this fact
Regards
This manuscript is a resubmission of an earlier submission. The following is a list of the peer review reports and author responses from that submission.